# Conservation-Compatible Retrofit Solutions in Historic Buildings: An Integrated Approach

Alessia Buda [1], Ernst Jan de Place Hansen [2], Alexander Rieser [3], Emanuela Giancola [4], Valeria Natalina Pracchi [1], Sara Mauri [1], Valentina Marincioni [5], Virginia Gori [5], Kalliopi Fouseki [5], Cristina S. Polo López [6], Alessandro Lo Faro [7], Aitziber Egusquiza [8], Franziska Haas [9], Eleonora Leonardi [9] and Daniel Herrera-Avellanosa [9],*

1 Department of Architecture, Built Environment and Construction Engineering (ABC), Politecnico di Milano, 20133 Milano, Italy; alessia.buda@polimi.it (A.B.); valeria.pracchi@polimi.it (V.N.P.); sara.mauri@polimi.it (S.M.)
2 Department of the Built Environment, Aalborg University (AAU), 2450 Copenhagen, Denmark; deplace@build.aau.dk
3 Department of Energy-Efficient Buildings, University of Innsbruck, 6020 Innsbruck, Austria; alexander.rieser@uibk.ac.at
4 Department of Energy, Centro de Investigaciones Energéticas, Medioambientales y Tecnológica (CIEMAT), 28040 Madrid, Spain; emanuela.giancola@ciemat.es
5 Bartlett School of Environment, Energy and Resources (BSEER), University College London, London WC1H 0NN, UK; v.marincioni@ucl.ac.uk (V.M.); virginia.gori@ucl.ac.uk (V.G.); kalliopi.fouseki@ucl.ac.uk (K.F.)
6 Department for Environment Construction and Design (DACD), Institute for Applied Sustainability to the Built Environment (ISAAC), University of Applied Sciences and Arts of Southern Switzerland (SUPSI), 6850 Mendrisio, Switzerland; cristina.polo@supsi.ch
7 Department of Civil Engineering and Architecture, Università di Catania, 95125 Catania, Italy; alessandro.lofaro@unict.it
8 TECNALIA, Basque Research and Technology Alliance (BRTA), 48160 Derio, Spain; aitziber.egusquiza@tecnalia.com
9 Eurac Research, Institute for Renewable Energy, 39100 Bolzano, Italy; franziska.haas@eurac.edu (F.H.); eleonora.leonardi@eurac.edu (E.L.)
* Correspondence: daniel.herrera@eurac.edu

**Abstract:** Historic, listed, or unlisted, buildings account for 30% of the European building stock. Since they are complex systems of cultural, architectural, and identity value, they need particular attention to ensure that they are preserved, used, and managed over time in a sustainable way. This implies a demand for retrofit solutions able to improve indoor thermal conditions while reducing the use of energy sources and preserving the heritage significance. Often, however, the choice and implementation of retrofit solutions in historic buildings is limited by socio-technical barriers (regulations, lack of knowledge on the hygrothermal behaviour of built heritage, economic viability, etc.). This paper presents the approach devised in the IEA-SHC Task 59 project (Renovating Historic Buildings Towards Zero Energy) to support decision makers in selecting retrofit solutions, in accordance with the provision of the EN 16883:2017 standard. In particular, the method followed by the project partners to gather and assess compatible solutions for historic buildings retrofitting is presented. It focuses on best practices for walls, windows, HVAC systems, and solar technologies. This work demonstrates that well-balanced retrofit solutions can exist and can be evaluated case-by-case through detailed assessment criteria. As a main result, the paper encourages decision makers to opt for tailored energy retrofit to solve the conflict between conservation and energy performance requirements.

**Keywords:** compatible retrofit solutions; built heritage; energy efficiency; sustainable preservation

## 1. Introduction

According to the United Nations Environment Programme (UNEP) [1], existing European buildings consume about 40% of the total energy consumption in Europe. For this

reason, in the last decades, several energy policies have been directed to deep renovation of the existing stock (as last 2018/844) [2].

Considering that more than one quarter of all European buildings were constructed before the 1950s [3], we can assume that many of them are of cultural, architectural, social and heritage values [4], hence in need of special attention for conservation purposes [5].

As preservation means maintaining the integrity, identity, and functional efficiency of a cultural asset [6], the renovation process may be an opportunity to improve the active functionality and avoid the decay of our built heritage. Tailored retrofit solutions (also referred to as "measures" or "interventions" in the text below) may improve the building conservation while acting on users' comfort and reducing the energy demand, which are crucial to ensure the continued use of buildings over time and consequently their endurance. In addition, and more crucially, by preserving the material fabric, the built heritage values are sustained. This is important to achieve holistic sustainable developments [7].

In stark contrast, projects mainly addressing environmental sustainability focus almost exclusively on measures enabling energy efficiency and cost savings, which may not be necessarily compatible with heritage values preservation [8]. It may turn out challenging for designers and practitioners to preserve cultural built heritage values while implementing retrofit solutions aiming at energy consumption reduction to achieve the key targets set in the 2030 Climate & Energy Framework [9]. Indeed, some authors have identified risks of destruction or significant impairment of some buildings' inherent heritage values if energy efficiency measures are implemented in isolation, verifying only the energy savings [10].

The number of publications addressing ways to improve the energy efficiency of buildings with cultural and architectural value recognized by the users has been constantly increasing during the last decade [11–13]. The balance between energy consumption reduction and conservation principles was the dominant criterion in recent literature [14], due to a pressing need to conserve the physical integrity of historic buildings.

However, the variability of historic constructions does not make it possible to identify in the literature retrofit strategies that can be considered exemplars applicable to all buildings. As indicated in several studies [14,15], professionals, and users, i.e., the building owners or the building occupants, emphasize the need for support during the decision-making process, as well as means to share best practices and repositories of retrofit solutions deemed suitable for the built heritage [16].

## 1.1. IEA-SHC Task 59: A Collaborative Research Project

In 2017, the International Energy Agency (IEA) has made the built heritage the focus of a new collaborative research project [17]. Within the Solar Heating and Cooling programme (SHC), 25 organisations (including public and private research institutions, heritage authorities, public administration, and industry) from 13 countries have joined forces in the IEA-SHC Task 59/Annex 76 "Deep renovation of historic buildings towards lowest possible energy demand and $CO_2$ emission (nearly Zero Energy Buildings-nZEB)" [18].

As Shah put it [19], "collaboration is used for solving problems that are too difficult or complex for an individual" (p. 216), and the identification of retrofit solutions that are compatible with historic buildings (and the barriers that prevent their implementation), is without a doubt an intricate task that profits from a multidisciplinary approach. The work carried out in IEA-SHC Task 59 relied heavily on knowledge exchange and task sharing, profiting from the wide and varied group of experts collaborating in the project. Thus, the methods used in this study were based on an iterative process of information seeking, comparing and synthesizing, making decisions, and finally making use of the synthesized solution [20].

This paper presents the work carried out within the IEA-SHC Task 59 project to support decision-makers in the adoption of conservation-compatible retrofit solutions for historic buildings, as suggested by the EN 16883:2017 [21] standard. The main objective of the study was to identify and evaluate examples that both satisfy the conservation of historic buildings and lower their energy demand.

The paper is structured in two main sections, capitalising on the work of the IEA-SHC Task 59 project. The first part introduces the major barriers towards the implementation of retrofit measures, looking at the role that legislation and economic viability, decision making approaches, and technical compatibility play in the planning process. The results presented here are the outcome of a collaborative literature review (or collaborative information retrieval [19,22]). The literature review was based on the recent developments in academic literature, with a focus on research that can be applied to historic buildings and complemented by grey literature published by heritage organisations and policy makers worldwide. The first part of this review, focused on the motivations and limitations of decision makers for the energy retrofit of historic buildings, was presented in [23]. In this occasion, however, the results looked specifically into the adoption of single retrofit solutions trying to identify the reasons that persuade decision makers to choose (or disregard) a certain solution.

The second section of this paper presents the IEA-SHC Task 59 approach to overcome some of these barriers. Firstly, the method adopted by the project members for the identification and documentation of conservation compatible retrofit solutions is presented. In this case, instead of a systematic review of scientific literature (as done for instance in [13]), the 73 experts involved in the project collaborated in gathering examples of compatible solutions implemented in retrofits of historic buildings or tested in research projects. This allows taking the review one step further linking the examples gathered to the actual cases where they were implemented. At this point, it is worth emphasizing that the expert group was not only formed by scholars but included also industry partners and members of heritage authorities and the public administration. Collaboration between academics and practitioners facilitates "knowledge co-creation" [24] and in this case ensured a multidisciplinary assessment of the solutions.

Organised in four working groups (walls, windows, HVAC systems, solar technologies) the expert members documented the identified solutions following a common template (further information is provided in Section 3.3). The assessment of these solutions is based on the criteria listed in the EN 16883:2017 standard. These criteria, however, have been revised by the different working groups to better adapt to the characteristics of each technology in a collaborative process of comparing, synthesizing, and making decisions. Finally, the webtool developed to gather and present these solutions to the end users is briefly presented. The details of this tool will be discussed separately in a future publication.

## 2. Drivers and Barriers When Implementing Retrofit Solutions in the Built Heritage

Many of the limitations preventing people from the energy retrofit in historic buildings are neither purely social nor purely technical, but rather the combined result of socio-technical issues [23–25].

Consequently, four clusters are proposed to group these barriers, as described in the following: (i) lack of confidence of decision makers in adopting technical solutions due to energy performance legislation requirements; (ii) lack of users' engagement in the retrofitting of historic buildings due to the reduced economic viability; (iii) lack of support and guidance in the retrofit design process for historic buildings, often too complex for non-specialised professionals and owners; and (iv) limited access to documented conservation-compatible retrofit measures that can ensure heritage compatibility and long-term performance.

This paper addresses the four points (Sections 2.1–2.4), and then specifically focuses on the fourth in the next paragraph.

### 2.1. The Impact of Legislation in the Adoption of Technical Retrofit Solutions

Cultural built heritage conservation is a cross-disciplinary field that is affected by numerous legal regulations at regional, national, and European level. These regulations address, among others, issues related to the safety, accessibility needs, functionality, comfort,

etc., of historic buildings. However, the lack of specific policies to support the improvement of energy performance is one of the main barriers in the retrofit planning phase [15,26].

Since the Energy Performance Building Directive (EPBD) 2002/91/EU [27] was published, European and national Directives have pushed existing buildings to become gradually more self-sufficient, focusing on cost-effective deep renovations leading to high energy performance. Deep renovation entails the application of energy efficiency measures able to transform existing buildings into nZEB [2]. This implies not only retrofit measures on the building envelope (e.g., thermal insulation, high energy performance windows) [28–31] and technical installations (e.g., advanced heating and cooling systems, and LED lighting) [32–34], but also the use of renewable energy sources [35].

However, the implementation of retrofit solutions that would be needed to reach minimum energy performance requirements set in building legislation may be detrimental to cultural built heritage values, especially in cases with high level of legal protection [8, 26,36,37]. For this reason, exceptions on the applicability of energy savings policies are provided in European countries legislation, if it is recognized that the retrofit "would unacceptably alter heritage building character or appearance" (art. 4) [2,38].

The existing gap between legislative barriers, regulatory exemptions and built heritage protection needs has made the selection and assessment of retrofit solutions an open question [15]. Refurbishment interventions adopted in buildings that are not perceived of heritage value are difficult to replicate in historic buildings where architectural and cultural values must be retained. However, as highlighted in many studies [39–41], a trans-disciplinary approach for the retrofit of historic buildings is possible. Solutions that improve energy efficiency and protect heritage values are available, although they need to be widely publicised and tailored on a case-by-case basis.

### 2.2. The Role of Economics in the Appraisal of Technical Retrofit Solutions

Economic viability of retrofit interventions may affect users' engagement with appropriate technical solutions [42–45].

In the case of public constructions, end users and real estate managers could be demotivated to act due to the high cost, complexity, and economic risks often associated with cases of complete renovation of an historic building [46]. A bias is recognized in the literature among institutional investors more familiar and comfortable with supply-side investments and large-scale financing (e.g., from EU project funding), rather than generally smaller projects on the demand side [47]. Indeed, it can be exceedingly difficult to secure a financial contribution for retrofit solutions that mix energy and preservation aspects as, for example, an improvement of indoor climate is not easy to be estimated in economic terms [48].

Lack of finance is often recognized as the main barrier in private residential properties, as building owners or occupants do not have access to enough funds for retrofitting. In fact, retrofitting requires upfront costs, and the benefits accrue gradually over time, often resulting in long payback periods [14].

When defining interventions, another driving factor in the decision making is linked to the difference between actual and predicted energy savings [49]. In general, stakeholders prefer the installation of retrofit products subsidized by national incentives, such as changing windows, or retrofitting the external building envelope with the installation of insulation materials [50,51]. However, the limitation of this approach lies in the reduction of the complexity of the planning intervention to one single aspect, without guaranteeing the preservation of historic building values as a whole—which is the scope of IEA-SHC Task 59.

### 2.3. The Complexity of Decision Making as Part of a Multidisciplinary Approach

Owners and building professionals involved in the retrofit process may be driven by different values (e.g., heritage and cultural, affective, aesthetic, architectural) in the decision-making process, potentially resulting in tensions—or synergies—towards the

identification of possible energy-efficient retrofitting solutions [7]. For example, while some people may value the aesthetics of specific features of the building (such as frescos, paintings, mosaic flooring, and decorated ceilings), others may value more the intrinsic historic aspect as an invaluable source of information on the past. The different attitudes may result in different hierarchy of priorities and, ultimately, acceptable solutions [4].

Moreover, in many cases none of the professionals involved in the retrofit are experts in energy efficiency of historic buildings, introducing a coordination and decisional challenge [48].

In projects where the contextual assessment of many aspects is required, guidance tools (i.e., web tools and decision support system), multicriteria analysis and toolkits can be used to support the complex decision-making process [52–54]. Several methods for the evaluation of interventions on historic buildings have been proposed and applied in current literature [55–57]. These methods allow the comparison of alternative scenarios during the decision-making process, based on a set of suitably selected evaluation criteria. However, many of them may be based on assumptions not referring to the values of the residents' assets [4].

In recent years, attention has been given to guidelines on the energy efficiency of historic buildings, written to support the adoption of an approach that integrates heritage values into energy efficiency plans. At European level, the EN 16883:2017 standard "Conservation of cultural heritage-Guidelines for improving the energy performance of historic buildings" describes the procedure for selecting appropriate interventions to improve the energy performance of a given historic building [21].

The EN standard recommends collaboration between owner(s), professionals, and heritage authorities, and clarifies that renovations or repairs are considered "conservation action only if it respects heritage significance and is based on evidence" (p. 10). In fact, it is exceptionally rare for residents to consult heritage agencies before repairing or retrofitting their dwellings [15,26].

More importantly, "heritage values" are based on assumptions that reflect the perspectives of heritage professionals rather than evidence of users' attitudes [7]. This is probably because studies relating to energy efficiency in historic buildings have mainly focused on the development of technical solutions [58–61]. Since built heritage values are perceived by professionals as a non-negotiable preliminary condition, the attitudes towards energy efficiency of people living in historic building may have been underestimated [4].

The dialogue between all stakeholders, especially in the early stage of the decision-making process, is of fundamental importance for defining the appropriate retrofit solutions [62,63] and should be encouraged. This might include knowledge gained from previous experience. However, a systematic collection of feedback from past projects seldom takes place, undermined by recurrent criticalities, such as difficulties in systematically retaining the knowledge acquired between projects, and impossibility of undertaking follow ups [23].

### 2.4. The Complex Nature of Conservation-Compatible Retrofit Solutions

One of the difficulties for professionals when selecting retrofit solutions is to understand the potential impact of their choices on the peculiarities of historic buildings in comparison to modern constructions [64]. This is because historic buildings relied on local resources and traditional construction techniques to ensure sufficient thermal comfort, lighting, and natural ventilation (e.g., stone, brick, or mixed masonry with high thickness ensuring a certain thermal insulation; windows and wall openings guaranteeing natural light and natural ventilation), with simple active strategies (e.g., fireplaces) to provide a suitable indoor environment [65–67].

Planning retrofit interventions therefore is a complex process of choosing solutions tailored to the specific case [68]. Thus, several authors in the literature have recognised the need to move away from the concept of Best Available Technology (from an energy efficiency point of view) and adopt the notion of Available Best Technology [40,41,69,70].

The latter acknowledges that an "improvement" may also be represented by an intervention within those available for the individual case in object, even though it may result in not meeting all the criteria required by law.

When improving the energy performance of the building envelope, such as roofs, walls, or windows, this needs a more comprehensive analysis to balance built heritage conservation, technical compatibility, health and comfort of occupants and energy efficiency. Nevertheless, there is no lack of solutions [28,71]; for example, capillary-active and reversible insulation systems tested on historic buildings are present in the literature [72–75].

Solutions for improving the airtightness of historic windows have been the object of several research projects [76,77], as well as solutions for improving the performance of historic windows with reduced impact on the character of the building [78,79] including shading systems [80,81].

In some cases, it may be more effective to focus on building services (HVAC systems) to obtain a significant reduction in energy usage by exploiting the latest technologies while minimizing the impact of retrofit interventions on the building envelope [41,82]. For example, the installation of a new heat pump system combined with the use of green energy, reusing the existing distribution system, might be a suitable measure to achieve a higher energy efficiency compared to traditional air-source heat pumps [29,83].

Implementation of other solutions, like a new heating system might require extensive intervention in the existing construction, including new heat distribution ducts that may be complicated or not possible to deploy due to limited space [84]. For this reason, some studies promote heating through small pipes mounted in or on the inside of the walls [85], or floor carpet heating systems aiming at reducing the spatial and material impact on heritage buildings [86].

The use of renewable energy systems in historic buildings, such as solar thermal collectors and photovoltaic systems, is often considered challenging as the common installation on the roof may alter the visual appearance of the building [87,88]. Nevertheless, it is possible to install solar thermal and photovoltaic systems (integrated or not) at historical buildings that fulfil the requirements of monumental protection, as demonstrated in several best practices documented in the [16,35,89,90].

The method/tool used to evaluate the energy performance of different scenarios may constitute an additional technical barrier for professionals in selecting suitable solutions.

Building energy performance simulations used by building professionals often overlook the hygrothermal behaviour of historic buildings as different to (modelling of) modern buildings. Although dynamic simulations are more complex to use and require more data than static simulations, they should be preferred as they best capture heat and mass transfer mechanisms predominant in historic buildings (e.g., thermal storage, and natural ventilation) [64,91]. The challenge in the use of dynamic simulations for built heritage is often associated to the lack of information on the hygrothermal properties (and their variability) of historic building materials and elements [61,92], leading to large uncertainties in simulation estimates if not accompanied by on-site measurements [58]. Further challenges are the provision of information on indoor comfort and indoor air quality necessary for a correct re-functionalization of spaces [93,94], and the lack of guidelines on the use of simulation tools in the context of historic buildings, hindering the reliability of the analysis outputs.

In general, the use of energy simulation tools as the only assessment method to evaluate retrofit scenarios for historic constructions does not allow a complete evaluation of solutions compatibility [64]. A successful method for evaluating retrofit solutions should, in fact, help to understand how the integration of different interventions can improve both the functionality (i.e., environmental, and microclimatic aspects) and the management of the building (i.e., costs and energy consumptions), preserving the heritage values of the latter [68].

Despite selecting compatible and well-calibrated solutions is considered a challenge for many, several tested products are present in the literature—although not always available to all.

The need to systematize solutions deemed suitable for historic buildings is accompanied by that of defining an integrated, adaptable, and consistent evaluation method that supports a whole-building assessment of scenarios, with the objectives of energy saving, environment, indoor quality, economic saving, and conservation.

## 3. The Challenge of Identifying Replicable Conservation-Compatible Retrofit Solutions

### 3.1. A Whole Building Approach in the IEA-SHC Task 59 Project

As discussed in the Sections 2.1–2.4, the different aspects of the IEA-SHC Task 59 project shall be seen in the broader context of the sustainable improvement of historic buildings. In this sense, a "whole building approach" is necessary, meant as an "integrated" approach that can maximize the strengths of different disciplines.

For this reason, the IEA-SHC Task 59 project has gathered a solid knowledge base on how to cost-effectively save energy in the retrofit of historic and protected buildings, thanks to the existing research and new findings shared by the partners involved in this interdisciplinary collaboration. The new approach developed to change the negotiation space of suitable retrofit measures was presented in a paper resulting from the IEA-SHC Task 59 [23] (Figure 1).

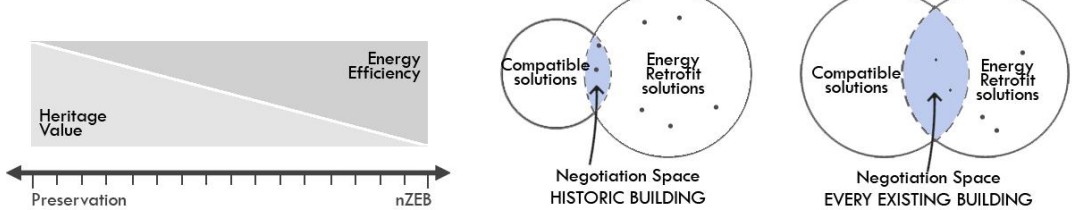

**Figure 1.** Negotiation space to select retrofit solutions in historic buildings. Data from [23].

For existing buildings without any heritage values to be considered during the renovation process, the choice of suitable solutions is much more extensive than for historic buildings. For the latter instead, the negotiation space includes all interventions that are considered compatible with the building characteristics and it strongly depends on the interaction of the involved stakeholders. The integration of all compatible solutions in this negotiation space would result in the lowest possible energy demand of the building.

The concept of "lowest possible energy demand" introduced in [23] acknowledges that in historic buildings the preservation of the heritage building value may sometimes result in absolute constraints on certain interventions. Similarly, it also spans a space from the concept of reducing energy demand close to the nZEB standard but do this with a focus on preserving as much as possible of the buildings aesthetic value to that of reducing the energy demand as much as possible while preserving all the buildings heritage values. Reality will lie in between, depending on the value of the building, and it will also consider additional parameters like comfort and economic feasibility [23].

### 3.2. Towards a Sustainable Approach in the EN 16883:2017 Standard

The European Committee for Standardisation has developed a suitable procedure to improve the energy performance of historical buildings, detailed in the EN 16883:2017 [21] standard "Conservation of cultural heritage-Guidelines for improving the energy performance of historic buildings". The guidelines are meant to be used by building owners, practitioners, and public sector to select appropriate solutions in the planning stage.

The procedure (Figure 2) helps in the selection of interventions, based on investigation, analysis and documentation of the building including its heritage significance. Rather than specifying general solutions beforehand, the EN 16883:2017 provides a procedure to facilitate the best decision for each individual building. The main goal is to find a

sustainable equilibrium between the use of the building, its energy performance, and its conservation.

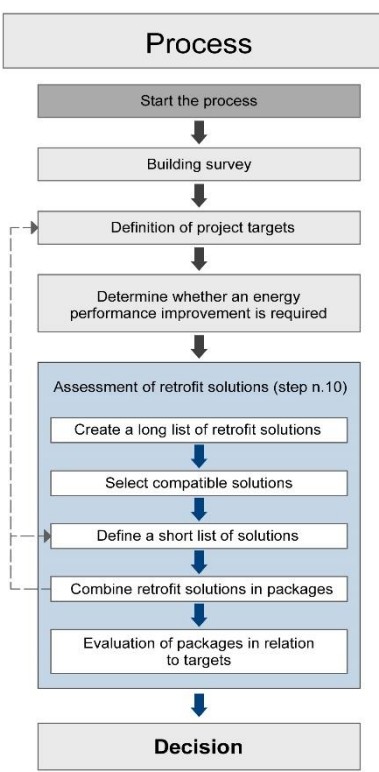

**Figure 2.** Flow chart of the procedure proposed in IEA-SHC Task 59 based on the EN 16883:2017 standard [21]. The light-blue rectangle indicates the working area for the Conservation Compatible Retrofit Solutions and Strategies activity [18] (p. 16).

Since the standard was intentionally kept very general, IEA-SHC Task 59 has been working on further developing this procedure with a focus in two areas:

- Multidisciplinary Planning Process (Subtask B): The scope of this thrust of the project is to increase the use and usability of the European guidelines to improve the energy performance of historic buildings and support professionals in the decision-making process. The main outcome is a handbook for the planning of energy retrofits in historic buildings.
- Conservation Compatible Retrofit Solutions and Strategies (Subtask C): This part of the project aims at further developing the assessment criteria (corresponding to step number 10) in the EN 16883:2017 that support the selection of solutions for energy refurbishment (Figure 2). The activity carried out in IEA-SHC Task 59 first led to the compilation of a long list of retrofit possibilities, identified from the information gathered through examples of good practice and research shared by experts and affiliated partners. Below, the general evaluation criteria available in the EN standard have been analysed and refined to make them specific to each building-element category of intervention and support the definition of a short list of solutions.

### 3.3. Conservation Compatible Retrofit Solutions and Strategies

One of the main objectives of the IEA-SHC Task 59 project was to identify, document, and assess replicable renovation solutions from different sources such as case studies, research projects (past and present) and innovative products from industry. The scope was making available to a larger audience a collection of already existing technical solutions to overcome some of the identified barriers (see Section 2).

As it has been anticipated in Section 1.1, IEA-SHC Task 59 is a task-sharing project that benefits from a large international network of researchers and practitioners working in the field of sustainability and heritage.

A collaborative information seeking methodology was adopted to collect retrofit solutions [19,22]. The IEA-SHC Task 59 members worked collaboratively on the shared task of collecting and organising retrofit solutions considered compatible for historic buildings. The collection of retrofit solutions was compiled from case studies gathered and reviewed by this large network of experts.

Partners also had the opportunity to provide information about retrofit solutions they had been working on in the past, for their validation and assessment under a common framework. As such, the uniqueness of this exercise is that it drew on the valuable source of information and long-term experience of the project participants.

A standardised procedure was defined to ensure that consistent information was collected for each case study submitted, and that all solutions included in the appraisal complied with the overall aim of the IEA-SHC Task 59.

These solutions should fulfil three main points:

- conservation compatibility with historic buildings,
- energy efficiency goals towards lowest possible energy demand and $CO_2$ emissions (nZEB),
- technical compatibility and functionality.

According to the EPBD Recast 2010/31/EU [38], nZEB is achieved when an energy balance is reached by bringing together architectural design, energy efficiency and local use of renewables. The goal of IEA-SHC Task 59, however, is not to present the equalized balance as absolute threshold, but as the intended target with the above threefold approach.

Specifically, the solutions had to demonstrate an increase in the energy efficiency of the building towards its lowest possible energy demand while ensuring their technical performance and the protection of heritage values.

To tackle the limited access to tested retrofit solutions that can ensure heritage compatibility and long-term performance, IEA-SHC Task 59 focused the review of retrofit measures compatible for historic buildings, on the following thematic areas:

- wall solutions: Thermal enhancement of external walls.
- window solutions: Conservation and restoration of historic windows with enhanced energy efficiency and user comfort.
- heating, ventilation, and air-conditioning (HVAC) systems: Ventilation systems and technical conditioning installations compatible for historic buildings.
- Solar technologies: thermal or photovoltaic systems, integrated or not, for historic buildings.

Every solution (Table 1) was documented following a common template. The first section aimed at collecting a brief overview of the solution with a description of the proposed intervention. Solutions included short information about the building context, pictures, and technical details.

**Table 1.** Number of documented solutions for each category in the IEA-SHC Task 59 Project.

| Building Component Category | Number of Documented Solutions |
|---|---|
| Walls | 37 |
| Windows | 16 |
| HVAC | 41 |
| Solar thermal collector or photovoltaic systems | 37 |

The next, and most important section, focused on the justification of why that solution would be feasible for the retrofit of a historic building from an energy, conservation, and technical point of view. A description of the case study where the solution was implemented

was subsequently collected, as well as any link (if available) to existing information and publications [23] to complement the documentation.

A set of 131 solutions has been documented so far, most of which with a link to real case studies of retrofitted historic buildings in Europe. This set is thought to be easily implemented as more solutions are available. Considering how much the context of a retrofit can change, this set of solutions is thought to be further adapted to the specific building and its location during the retrofit design process.

A parallel aim of the IEA-SHC Task 59 project was to propose a list of criteria to assess the suitability of the solution when applied to a specific historic building. For this purpose, the risk criteria listed in the EN 16883:2017 standard were adopted in this work as starting point.

In the EN standard, the definition of these risk criteria is based on a risk–benefit scheme and considers: *technical compatibility* (e.g., hygrothermal risk, structural risk, or corrosion risk), *heritage significance of the building* (estimated as visual, spatial, and physical impact on the heritage), *economic viability* (cost–benefit evaluation), *energy* (primary energy demand), *environment* (sustainability of products), *indoor environmental quality* (thermal comfort levels and air quality assessment), *impact on the outdoor environment* (impact on the building context) and *aspects of use* (impact on building management issues).

These criteria, however, are not solution-specific and their application is not always immediate. Within the IEA-SHC Task 59 project, partners were organised in working groups to refine and adapt these criteria in support of the assessment of retrofit solutions for the different building-element categories explored: walls, windows, HVAC systems, and solar technologies. More information on the different categories and how the assessment was carried out is presented in the following sections.

### 3.3.1. Walls Solutions

The collection of solutions for improving the energy efficiency of external walls was clustered in five categories with different characteristics and impact on the historic building integrity: (i) reversible systems; (ii) internal wall insulation; (iii) cavity insulation (behind internal lining); (iv) frame infill insulation; and (v) external wall insulation.

This collection is characterized by a wide variety of materials: mineral wool, cellulose, wood fibre, cork, calcium silicate, perlite, aerogel, phenolic foam, rigid polyurethane (PUR), etc. The insulation materials selected also present different behaviour and characteristics (e.g., insulation systems with vapour control layers, and capillary-active systems).

Of the 37 solutions documented, two solutions are considered reversible systems: one solution for a reversible façade element installed on the outside, and one of a thin wooden panelling using straw insulation.

Sixteen examples of internal insulation are documented. They are divided between solutions with capillary active insulation materials (12 solutions, such as perlite panel, or wet blown cellulose applied between frames); solutions with vapour retarder (4 solutions, using insulation materials such as mineral wool, cellulose and sheep wool).

Four different solutions of cavity insulation behind internal lining have been documented, including blow in materials, like aerogel-based material, cellulose, and injected foam insulations.

Examples of external insulations are also documented in the collection (nine solutions), including solutions with mineral wool (four solutions), with vacuum insulation panels (one solution), and in combination with thin internal insulation (reed mat, multiport, and wood fibre) (four solutions).

The collection also includes two examples of frame infill insulation, one of a half-timbered building with hemp concrete and another one with timber walls and wood fibre.

Advanced solutions still under development are also included in the collection, like reflective coating, aerogel-based textile wallpaper, or reversible external façade systems. For these special and innovative solutions, the related documentation has been reviewed by the IEA-SHC Task 59 members from scientific and grey literature papers.

Twenty-four solutions of the whole set were implemented on a historic building and are documented as best practices; five of them have even been further assessed with simulations and on-site measurements.

General topics such as driving rain protection and wall drying are also documented in the majority of the collected examples.

The assessment of these solutions in the specific case studies was carried out by adapting the criteria in EN 16883:2017 standard (listed in Section 3.3) to the requirements for walls. Specifically, the criteria recommended by the EN standard were extended by tailoring the description to walls of historic buildings and should thus facilitate practical application. This aspect are discussed more in detail in a companion paper within the IEA-SHC Task 59 frame of activities [95].

### 3.3.2. Windows Solutions

Two main criteria were considered for the collection of window solutions. The first criterion concerns the identification of the most common historic window types. Four types were pinpointed: single window, single window with winter window, coupled window, and box-type window. The interventions applicable to these types of historic windows were then grouped into four levels of increasing impact on the character and visual appearance of the building, which is the second criterion considered.

The windows solutions documented are divided into (i) low-impact interventions (four solutions), i.e., conservative options potentially applicable to any window with no visual, material or spatial impact on the historic building (e.g., inserting a sealant, repairing, etc.); (ii) interventions with impact on the inside (six solutions), i.e., addition or substitution of single window elements with limited impact on the building character and appearance from the internal side of the façade (e.g., installing an internal new window layer); (iii) interventions with impact on the outside (four solutions), i.e., addition or substitution of single window elements with limited impact on the building character and appearance from the external side of the façade (e.g., replacing outer glass, installing an external new window layer); (iv) strategies with a profound impact on the building character and appearance (two solution), i.e., replacement of the window with a new component. The solutions revised not only concerned interventions on the window components (i.e., frame, and glass), but were also extended to shading systems (such as shutters, blinds, and curtains), which can have a strong influence on the thermal performance of the window itself.

The subdivision proposed makes it easier for the building owner to find a viable solution; if the elements to be preserved are limited to the façade, it will be possible to select interventions focused on changing only single elements with impact on the internal appearance and vice versa. For listed buildings, the available options might be limited to minimal interventions, like repairing original windows or improving their airtightness.

For 13 out of 16 solutions, at least one practical case study was collected. In some cases, a detailed energy assessment has also been carried out.

An assessment of window solutions adapting the criteria in the EN 16883:2017 standard was also developed as an output of this activity, which allows a systematic evaluation of different solutions.

### 3.3.3. HVAC Solutions

HVAC solutions focused on two main areas: ventilation systems, and heating/cooling systems. Solutions capable of producing positive effects on energy efficiency, indoor air quality and climate were considered. The review focused on minimal invasive solutions suitable to the future use of the building, while reducing risks for the built heritage conservation (e.g., moisture damage, especially in conjunction with internal wall insulation).

For ventilation systems, the documented solutions range from natural ventilation and space-saving (such as active overflow systems or the push pull system) to artificial ventilation solutions in historic buildings. Three documented solutions deal with the airtightness of buildings, a fundamental requirement for the installation of mechanical

ventilation systems. Five examples of central ventilation systems with suspended ceilings are also collected. There are also two examples where the distribution takes place through the floor construction.

Decentralised systems such as ventilation with monoblocks air handling units and room-by-room systems include four examples. Furthermore, one example of alternative possibilities (i.e., facade-integrated ventilation, air supply via chimney/shafts and active overflow systems) were documented. Thus, a total of 17 solutions for the integration of ventilation systems in historic buildings were collected.

For heating/cooling solutions they were collected examples with heat pumps (four solutions), pellet (two solutions), and wood chip boilers (one solution) and cogeneration system (four solutions). Examples of biogas and district heating were described and documented in connection with different distribution systems, like floor heating (four examples), wall heating (one example), and normal radiators (three examples). In addition to that, separate examples such as radiators with visible piping, air heating and infrared heating panels with general descriptions have been included in the collection.

In all the cases involving heating solutions, the integration of the distribution system may be particularly challenging. For this reason, different systems like conventional floor heating, wall heating, radiators, air heating, infrared heating panels, etc., were assessed. In total 18 ventilation solutions, 12 distribution solutions, and 13 production solutions were documented.

On the basis of the assessment criteria in the EN 16883:2017 standard, an approach for tailoring the selection of solutions on a case by case has been developed. A detailed description and an application of this assessment method for HVAC solutions are presented and discussed in companion paper within the IEA-SHC Task 59 framework [96].

### 3.3.4. Solar Technologies

New technical solutions with high-performance levels may allow an efficient use of solar energy while preserving the character, heritage and architectural quality of historic buildings and sites. The documented solar energy solutions (37) mainly concerned solar thermal collectors and photovoltaic systems compatible with historic buildings.

The collected case studies demonstrate that most solutions used to date in historic buildings are roof-integrated systems (22 solutions out of 37).

Pitched roofs and steep-roofed houses, widely built until the 20th century, are not only characterized by their shape and contours but also by the construction, nature and characteristic colours of the surface materials used (e.g., ceramic or slate tiles, copper or zinc roofs, etc.).

In seven case studies, the solar thermal and photovoltaic systems have the same colours as the roof and therefore were well concealed. In another three cases, systems are not visible from the street and sometimes they are just part of the architectural concept.

Four solutions describe systems attached to the roof, which are mostly not visible from the street and, therefore, may result more compatible in historic contexts (such as in historic city centres) due to the lower visual impact on the appearance of the historic building.

Three wall-integrated systems are also documented, with several interesting examples of their application to valuable historic building. These case studies demonstrate that a harmonization between conservation and renewable energy sources is possible. Alternative solutions with free standing systems are documented as well for cases where one of the solutions above may not be a possibility.

To complete the collection and to provide alternatives for special cases, four models for sharing renewables, i.e., shared solar energy projects for building complexes/communities, are also documented.

All solutions have been assessed adapting the criteria in the EN 16883:2017 standard to analyse their strengths and weakness.

### 3.4. A Decision-Support Tool for the Identification of Solutions

A tool named "HiBERtool" (Historic Building Energy Retrofit tool) is being jointly developed between the IEA-SHC Task 59 and the Interreg Alpine Space ATLAS project [18,97] to help end users (whether architects, engineers, or building owners) identifying a list of suitable solutions.

The webtool interface guides the user to a set of suitable solutions, depending on their needs and requirements. A decision tree was created for each of the different building components presented in this paper (walls, windows, HVAC systems, and solar technologies) (Figure 3). The trees should enable the user to narrow down all the solutions to those suitable for the specific case study (in correspondence with the assessment of solutions approach proposed in the EN 16883:2017 standard—Figure 2).

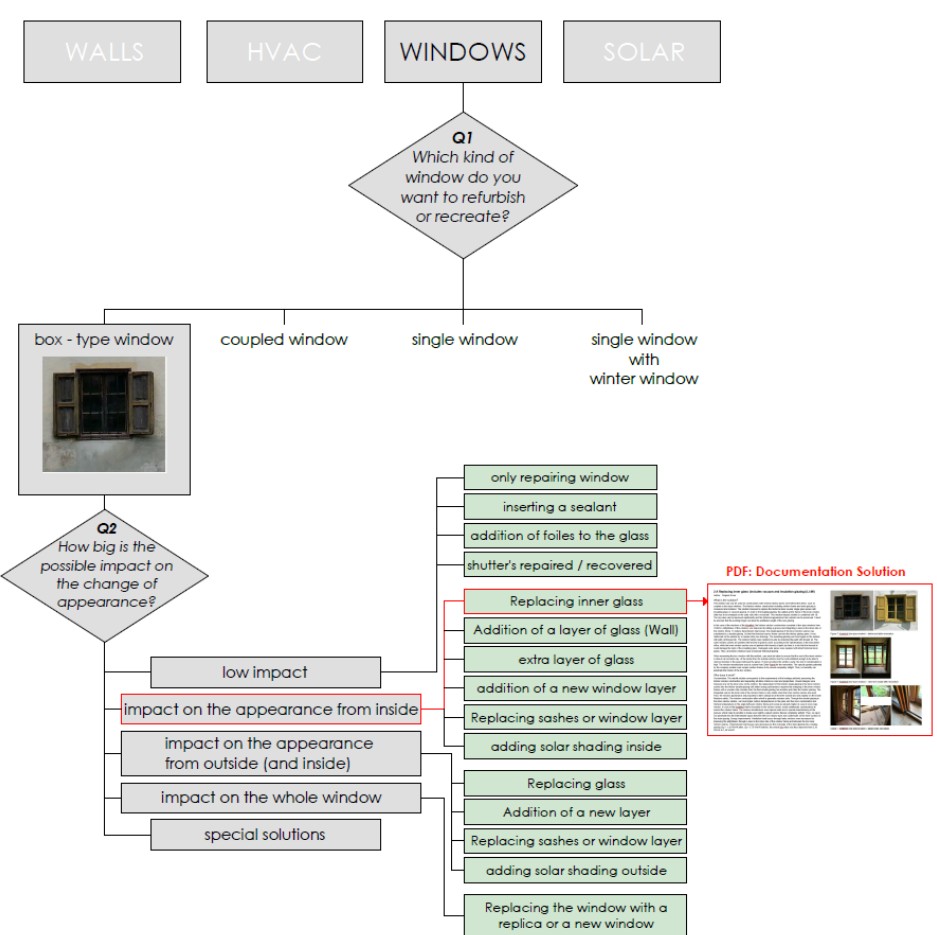

**Figure 3.** Application of the decision tree for windows in the HIBERtool.

In the online tool, retrofit solutions are selected from those available in the repository by answering simple questions about the building, its context, or the level of conservation.

Figure 3 illustrates one of this decision trees and how based on a series of simple questions the user is guided in the choice of a retrofit solution for an historic window. With the first question, a classification is made according to the type of element to be refurbished. The second level asks about the limits on the modification of the window. The number of questions asked depends on the category. For instance, the HVAC tree presents a series of up to five questions to identify the suitable solution. These choices lead in the decision tree to a list of possible measures. A detailed description of the selected solution can be saved as a PDF file.

The HiBERtool offers a comprehensive and structured access to implemented solutions. Furthermore, the solutions presented are linked to exemplary retrofits documented in the

HiBERatlas online platform (a separate output from the same projects) [16,98]. In contrast with general categories of solutions [99], practical tested solutions offer the advantage of the experience gained during their implementation.

The tool, with 131 solutions will thus serve as an inspiration and provide a useful basis for the planning process. In order to allow for the integration of future documentation into the tool, a structure was designed to be continuously expanded as desired.

## 4. Discussion and Conclusions

Numerous decisions arise when implementing energy retrofit solutions in historic buildings. Indeed, the gap between legislative requirements, regulatory exemptions and heritage protection needs has made the selection and evaluation of technical solutions a challenge. At the same time, the economic feasibility of technical solutions plays an important role in the assessment of solutions: the more profitable are more likely to be funded and more appealing for decision makers. However, technical solutions can also have an impact on other aspects, such as heritage conservation, indoor environment quality, $CO_2$ production, and use of the building.

The selection of technical solutions is therefore seen by the decision-makers of the project (e.g., practitioners, owners, and heritage authorities) as exceedingly difficult. This, together with a limited access to the existing solutions, is often preventing them from implementing compatible solutions in historic buildings.

Meanwhile, a wide range of balanced solutions exists; solutions that meet the need to improve the historic building sustainability, lowering the consumptions and respecting the building integrity.

The IEA-SHC Task 59 project results showed that it was possible to identify and assess numerous replicable renovation solutions for windows, walls, HVAC systems and solar technologies that have been proven in practice, in relation to conservation compatibility, energy efficiency and technical compatibility. One of the main results of this work is the large set of solutions identified by the IEA-SHC Task 59 members, coming from real case studies of retrofitted historic buildings all over Europe. The multidisciplinary nature of the IEA-SHC Task 59 research group allowed selecting and assessing solutions that met the goal of improving the energy performance of historic buildings while compatible with the building conservation, and to create a diversified set of solutions, that includes different techniques and technologies.

With the decision support tool HiBERtool, a new possibility is offered to explore and find different solutions for the energy-efficient retrofit of historical buildings. This tool enhances transferability ensuring that knowledge gained in a project is passed to future retrofits by making existing solutions accessible to others. While existing decision tools often end on a general level (e.g., internal insulation) [99], the HiBERtool documents the solutions on a product level (system used, material, installation, etc.) and linked to a case of practical implementation. The tool can also be easily expanded if more examples are available.

The results of the IEA-SHC Task 59 project also show the need to adapt the assessment criteria included in the EN 16883:2017 standard to better support the evaluation of retrofit solution and facilitate its practical application. Despite the efforts in the IEA-SHC Task 59 in adapting the assessment criteria, the evaluation of possible solutions remains a complex task as it requires special attention and evaluation of the elements that are to be preserved, the context in which the intervention is placed, and the initial project objectives. A successful implementation of retrofit solutions in historic buildings will always remain the output of a multidisciplinary decision-making where several stakeholders with different expertise and priorities are involved.

Nevertheless, the approach presented here offers a systematic assessment of possible measures that will facilitate the tailor-made renovation for historic buildings.

**Author Contributions:** Conceptualization, A.B., A.R. and D.H.-A.; Investigation, A.B., A.R. and D.H.-A.; Methodology, A.B., E.J.d.P.H., A.R. and D.H.-A.; Resources, A.B., E.J.d.P.H., A.R., V.M., V.G., C.S.P.L., S.M., A.E. and F.H.; Supervision, A.B. and D.H.-A.; Visualization, A.R.; Writing—original draft, A.B., E.J.d.P.H., A.R., E.G., V.M., V.G., C.S.P.L., A.L.F., A.E., E.L., F.H., V.N.P., S.M. and D.H.-A.; Writing—review and editing, A.B., E.J.d.P.H., A.R., E.G., V.G., K.F., A.L.F. and D.H.-A. All authors have read and agreed to the published version of the manuscript.

**Funding:** The authors thank the Department of Innovation, Research and University of the Autonomous Province of Bozen/Bolzano for covering the Open Access publication costs. The authors wish to express their gratitude to the IEA-SHC and EBC Executive Committees for supporting the Task59/Annex76. The authors are especially grateful for the financial support from the European Regional Development Fund under the Interreg Alpine Space programme to the Project ATLAS (ID: ASP644); the Regional Development Fund and coordinated by the Federal Office for Spatial Development (ARE) and the Swiss Federal Office of Energy (SFOE), contract no.: SI/501896-0 (Cristina Silvia Polo López); the Danish National Energy Technological Development and Demonstration program (EUDP) Grant 64017-05175 (Ernst Jan de Place Hansen); the UCL Bartlett Synergy Grant; the EPSRC Platform Grant EP/P022405/1 (Valentina Marincioni); the EPSRC UCL Doctoral Prize Fellowship Grant EP/N509577/1 (Virginia Gori).

**Acknowledgments:** The authors wish to thank all the experts in the IEA-SHC Task 59/Annex 76 for their valuable contributions.

**Conflicts of Interest:** The authors declare no conflict of interest. The funders had no role in the design of the study; in the collection, analyses, or interpretation of data; in the writing of the manuscript, or in the decision to publish the results.

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
