# Peer review of "Conservation-Compatible Retrofit Solutions in Historic Buildings: An Integrated Approach"

_sustainability, doi:10.3390/su13052927_

Round 1

Reviewer 1 Report

The article is presented as a research paper, but, is it certainly? Leaving aside its usefulness, is this article a high quality scientific work? (As the MDPI Publication Ethics Statement says), is it adequate for an advanced forum for studies related to sustainability (see Aims of the journal). I have great doubts.

This is a paper that, firstly, describes the approach devised in the IEA-SHC Task 59 project (Renovating Historic Buildings Towards Zero Energy), and secondly, presents a decisión-support tool for the identification of solutions, within the framework of the EN 16883:2017 standard.

It looks more a collection and analysis of solutions than a piece of research itself. It seems to confuse a method (a procedure followed in the sciences to find the truth and teach it) with a process (a set of the successive phases of an operation).

Regarding the tool developed between the IEA-SHC Task59 and the Interreg ALpine Space ATLAS Project, called HiBERtool, it should be said that there have already been applications and tools presenting the various interventions that can be carried out on building elements of historical buildings. An example is the Responsible Retrofit Guidance Wheel  (http://responsible-retrofit.org/wheel/), by The Sustainable Traditional Buildings Alliance (STBA), released in 2014. This Wheel has been designed to address such issues by clearly identifying different benefits and concerns, by referencing the most relevant and accurate information, and by providing a systemic and holistic approach to retrofit design, application and use. The Wheel is both an aid to decision making and a way of learning about traditional building retrofit.

In the case of the HiBERtool, it follows from its description (section 3.5) that some of the drivers and barriers listed in section 2 (the impact of legislation and the role of economics, mainly), have not been considered, so that half of the problem is omitted.

On the other hand, we have a large team that has participated in the development of a decision-support tool for the identification of solutions. The usefulness of the tool is based on setting the different ways of intervention; however, one by one, these types of intervention are not new. They are all invented.

Regarding the authorship of the article, I find an inconsistency in relation to the number authors. In fact, the large team that may be necessary for the development of this tool, looks excessive for a research paper, more so in this case. Have 9 researchers really been needed to conceptualize this article? 6 authors to establish the methodology?

Reviewer 2 Report

The manuscript concerns a methodologic approach to support decision-makers to individuate and assess retrofit solutions for existing built heritage, considering both listed and unlisted ones.

The research paper clearly describes the research scope, and interestingly refers to those barriers that make difficult to apply appropriate retrofit solutions to existing buildings. From this assumption, research is directed towards a framework for the individuation of compatible retrofit solutions, balancing between conservation and energy performance improvement aims.

The working area refers to an operative development of Guidelines for improving energy performance of historic buildings reported in EN 16883:2017. 

I suggest to deepen both methodology description and results discussion, in order to better fill the declared goal of the presented research. In detail:

1) Please provide a more detailed description of the process you followed up to individuate the set of considered retrofit solutions. In my opinion, it is very important to show how these 130 solutions represent a significant range, and in which scope: a) on one hand, EN 16883: refers to historic buildings, while your research covers a wider heritage; b) on the other hand, point out the contexts (geographical context, use of construction techniques contexts, etc.)  within this solutions' set is applicable.

2) Please show how solutions' set has been formed, explaining its structure and all classification criteria (sections 3.4.1 - 3.4.4.). For example, with reference to section 3.4.2., only 6 over 16 retrofit actions are generically depicted. It is important to highlight how much rigorously research has been conducted.

3) About HiBERtool decision support, Figure 3 shows an "Example" of a "generic decision tree": this sounds as an oxymoron. Once you provide the decision tree of the tool, it would be useful to show an application example, such as a case study used to testify the tool. At this manuscript stage, it is not clear at all how the solutions set has been implemented / embedded into HiBERtool: which relationship between walls, windows HVAC and RES solutions and decision trees? Please detail.

4) Moreover, a report on an application could strengthen your conclusions, with a critical discussion over strengths and weaknesses of both solutions' set and HiBERtool.

Reviewer 3 Report

the paper represents a robust state of the art of the activities of the IEA-SHC Task 59, exhibited in the context of international research, mainly of a European context, on the theme of Renovating Historic Buildings Towards Zero Energy.

I believe that the description of the technological solutions for the intervention on historic buildings is an interesting and complete framework of reference on the subject, perhaps too concise.

The paper represents the synthesis of the activities of a very extensive research program and certainly loses something in the extreme synthesis of the results of the activities, but it is nevertheless a very useful article also as a scientific dissemination on a topic in which in different national contexts it is very difficult to find common guidelines.

Round 2

Reviewer 1 Report

After reading the letter of the authors, the arguments put forward by them have been convincing. In addition, thanks to the improvements, the article has gained clarity and consistency.

Reviewer 2 Report

After the 2nd round review, I found out that criticalities that affected the manuscript at 1st round have been resolved.

I found out a clearer description of retrofit solutions set and a useful example that shows how HiBERTool works.